# Express Diagnostics of Proteolytic Activity of Periodontopathogens—Methodological Approach

**DOI:** 10.3390/dj10110217

**Published:** 2022-11-21

**Authors:** Ekaterina Aronova, Marina Dmitrienko, Anastasija Ivanova, Yulia Gaykova, Anna Kurochkina, Alisa Blinova, Julia Bazarnova, Elizaveta Paponova

**Affiliations:** 1Graduate School of Biotechnology and Food Science, Peter the Great St. Petersburg Polytechnic University, Polytechnicheskaya Street 29, 195251 Saint Petersburg, Russia; 2Association of Medicine and Analytics Co., Ltd., 17 line V.O. 4-6, 199034 Saint Petersburg, Russia; 3Department of Periodontology, Tver State Medical University, Sovietskaya Street 4, 170100 Tver, Russia

**Keywords:** proteolytic activity, oral cavity microbiota, periodontal disease, protease, periodontal pathogen, red complex, point-of-care testing

## Abstract

The species spectrum of the oral microbiome is considered to be the key factor in the development and progression of periodontal inflammatory disorders. The “red complex” including *Porphyromonas gingivalis*, *Tannerella forsythia* and *Treponema denticola* has the highest pathogenic potential. These bacteria have several biochemical mechanisms that allow them to colonize and destroy periodontal tissues. Proteolytic enzymes play a crucial role in this process. Early diagnosis of pathological conditions induced by microbial contamination allows for the timely treatment of periodontitis. Otherwise, the development of the disease may lead to tooth loss. A total of 48 patients aged 18 to 65 years old who required professional oral hygiene were recruited for this clinical study. Microbial content analysis of dental plaque from the interdental space and the back of the tongue was performed using real-time PCR. To determine the proteolytic activity of oral bacteria, the new express diagnostic method was applied (diagnostic sensitivity, 0.875; specificity, 0.928). The results demonstrate a strong and significant correlation between the new method and the PCR analysis (r = 0.785, *p* < 0.001). These results show that the new express method can be valuable as an early diagnostic method for periodontal inflammatory disorders caused by the “red complex” bacteria.

## 1. Introduction

Inflammatory periodontal diseases (PDs) are known as the most common dental disorders in the world. They are a medical, social and economic problem and also lead to a decreased quality of life [1]. In the past few decades, PDs have been registered more frequently in all age groups, including children and adolescents [2,3,4]. PDs affect more than 50% of adults, and 14% of them suffer from the severe forms of the disorders. By the age of 65, from 70 to 98% of the population are challenged with PDs [3,4,5,6,7]. Aggressive, recurrent PDs continue spreading [7]. More and more studies are proving the negative impact of periodontal pathogenic microorganisms and tissue inflammatory mediators on somatic health [6,7]. Periodontal disorders may be asymptomatic in the early stages of the disease, and patients usually ask for medical help because of pronounced signs of the PD: bleeding, swelling of the gums, dental mobility or tooth loss. It is important to timely detect and stop the inflammatory process in the dentogingival junction (gingivitis) that otherwise would spread and affect the epithelial and connective tissues of the dentogingival junction [7,8].

The main etiological factor of gingivitis and periodontitis is the microbial biofilm [9]. The tissue damage is based on the interaction between the aggressive microbial factors and the local immune reactions [10]. Microbial toxins and enzymes, intensive bacterial growth and local phagocytic responses lead to dentogingival junction pathology, chronic inflammation, destruction of the periodontium and the formation of periodontal pockets. Peri-implant inflammation is caused by this process too. The frequency of peri-implantitis is increasing due to the rising popularity of prosthetic rehabilitation with dental implants [11]. The long-term persistence of pathogenic bacteria induces implant failure at all stages of treatment [12]. Additionally, poor oral hygiene through the period of orthodontic treatment may facilitate the formation of dental plaque or tartar and lead to intense microbial colonization of the oral cavity. The commensal microbiome has an important role in the maintenance of oral and systemic health. Balancing interference causes oral pathologies such as cavities, endodontic disease, periodontal disease, osteitis and tonsillitis and can be associated with the development of several systemic diseases, such as cardiovascular disease, ictus, pre-term childbirth, diabetes, pneumonia, obesity, colon carcinoma and psychiatric issues [13].

The oral cavity microbiota is estimated to contain over 700 species of bacteria, fungi, viruses and protozoa [14]. The latest scientific discoveries prove that the pathogenesis model of PDs relates to disorders initiated by a dysbiotic, synergic microbiota [15]. The most common species are both Gram-negative (*Porphyromonas gingivalis*, *Treponema denticola*, *Prevotella intermedia*, *Aggregatibacter actinomycetemcomitans*, *Campylobacter rectus*, etc.) and Gram-positive bacteria (*Eubacterium timidum*, *Parvimonas micra*, etc.) [7,9]. Combinations of the most virulent pathogens are common among aggressive and destructive forms of PDs [16]. The pathogenicity of microflora to the periodontium is determined by proteolytic enzymes in the bacteria cell wall and the endotoxins that are directly involved in periodontal tissue destruction. The “red complex” bacteria (*P. gingivalis*, *T. forsythia* and *T. denticola*) are considered to be the most aggressive bacterial agglomeration [17,18]. Proteolytic enzymes are established to be the virulence factors of *P. gingivalis* and *T. denticola* [19]. *P. gingivalis* proteolytic enzymes degrade fibrinogen and destroy or transform cytokines such as tumor necrosis factor alpha (TNF-α), interferon gamma (IFN-γ), interleukin-6 (IL-6) and interleukin-8 (IL-8), and they also cleave or inactivate the C5a receptor of phagocytes, induce the chemotaxis of neutrophils and activate prothrombin, C-reactive protein and neutrophils [20].

The high frequency of PDs, their multifactorial etiology, difficulties in achieving stable remission, spreading of severe forms of PDs among young people and the low accessibility of periodontal care render the problem of early diagnostics urgent. At present, there are numerous methods for PD diagnostics: bacteriological analysis, the polymerase chain reaction (PCR) method, saliva gas–liquid chromatography and spectrophotometry, pH measurement and proteolytic activity evaluation [8].

The proteolytic activity of the “red complex” bacteria is assumed to be an important virulence factor; therefore, evaluation of the enzymatic activity of these bacteria allows for the prediction of inflammation in periodontal tissues at an early stage. Point-of-care testing for proteolytic activity offers many advantages as it is easily obtainable, sample collection is non-invasive and the procedure is performed in a doctor’s office. We hypothesized that express diagnostics of the proteolytic activity of periodontopathogens can help in determining the risk and severity of periodontal inflammation in periodontitis patients and can be used as a PD screening method. The aim of this study was to develop an innovative express methodology that allows for the estimation of the proteolytic activity of periodontopathogenic bacteria in dental and tongue plaque and to compare this method with molecular genetics methods, such as microbiological assays and saliva biochemical analysis.

## 2. Material and Methods

### 2.1. Patients

A total of 48 patients aged 18 to 65 years old (coded as d01–d48) who required professional oral hygiene were recruited for this clinical study. This study included patients with chronic gingivitis, chronic periodontitis and halitosis, as well as patients with soft and mineralized plaque without any complaints. Before the procedure, consent was obtained from every patient. The criteria for patient exclusion were: genetic diseases, acute and chronic somatic diseases in the decompensation stage, autoimmune diseases, acute and chronic infectious diseases and diseases of the oral mucosa.

The research was performed at the Department of Periodontology (Tver State Medical University). Firstly, the oral cavity examination was performed, and the Silness–Loe hygiene index was measured for all teeth [21]. PCR analysis of plaque in the interdental space (between 1.5 and 1.6) and at the back of the tongue, as well as the bacteria proteolytic activity express estimation, was conducted for each patient. Additionally, the saliva biochemical evaluation was performed on 10 randomly chosen samples.

### 2.2. Sampling Method

Tooth brushing, rinsing with mouthwash and food intake were not allowed 3–4 h before the sampling. Sterile dental applicators were used to perform sampling from the interdental space, in addition to scraping the back of the tongue. Material application on the segment for the express diagnostics was conducted immediately after the sampling. For the PCR analysis, the absorbers were placed into Eppendorf vials with a transport medium. For the saliva biochemical assay sampling, the patient had to sit with their head down and was instructed not to swallow the saliva and not to move their tongue and lips. It was necessary to accumulate the saliva in the oral cavity for two minutes and to spit it into a sterile container. This procedure was performed three times. In case of oral mucosa damage or bleeding gums, saliva analysis was prohibited.

### 2.3. Express Diagnostics of Proteolytic Activity in the Oral Cavity

Analysis of the patents over the past 30 years reveals multiple technologies for the express diagnostics of the proteolytic activity of bacteria in the oral cavity [22,23,24]. Express diagnostics is best studied in research aimed to determine the pathogenic bacteria and to diagnose PDs [23,24,25,26,27]. Express diagnostics of proteolytic activity is based on the colorimetric method [23]. Relying on information about the level of the technologies, the disposable device shown in Figure 1 was tested.

The device contains one chromogenic test substrate that is specific to the proteolytic enzyme that is released by the periodontopathogenic bacteria in the oral cavity, and another chromogenic test substrate that is non-specific to the microbial enzymes. The second chromogenic test substrate is degraded by the enzymes in response to the amount of bacteria. The concept of the device is based on chromophore formation as a result of lysis of the chromogenic substrate by the trypsin-like enzyme which is released by the “red complex” bacteria. The hydrolysis is accompanied by a color reaction, which indicates the presence of proteolytic activity in the sample taken from the oral cavity (Figure 2).

For the analysis to be performed, the biomaterials collected from the interdental space and the back of the tongue were simultaneously placed into the different parts of the chromogenic substrate layer of the device. Interpretation of the results was conducted 10 min after placing the device into the incubator with a temperature range of 38 to 45 °C. The result was interpreted as positive when the device indicator element changed its color from salmon to light blue or blue. This element is represented by the hydrophilic carrier that contains the chromogenic reagent. The stain color intensity was estimated with the help of a colorimetric scale (Figure 3) verified during the indicator element laboratory assessment with the use of trypsin solution at concentrations from 0.0167 mg/mL to 5 mg/mL. If there was no color change, the result was interpreted as negative.

### 2.4. Methods and Conditions of the Polymerase Chain Reaction Setup

Quantitative PCR analysis of dental and tongue plaque bacterial DNA was applied as a reference method. The DNA was extracted from samples with the “Prep-NK-Bio” kit for the DNA and RNA extraction. For the detection and quantitative assessment of periodontopathogenic bacterial DNA in the biological materials through PCR analysis with fluorescence hybridization in real time, the “Dentascrin” (“Liteh” manufacturer, Russia, One-Step-PB-60 set) kit was used. The amplification, analysis and interpretation of the results were performed according to the guidelines of the reagent kit.

For the DNA analysis, the following devices were used: a Neoteric Lamsystems laminar flow unit for the DNA extraction (Lamsystems, Miass, Russia), an Eppendorf centrifuge 5415R (Eppendorf, Hamburg, Germany), a BioSan TS-100 thermo-shaker (BioSan, Riga, Latvia), a BioSan Combispin FVL-2400N vortex (BioSan, Riga, Latvia) and a Visma Planar suction device (Visma Planar, Minsk, Belarus).

For the DNA extraction from biological materials, the “Prep-NK-Bio” kit lot E004 (OOO “Alkor Bio”) was used. The PCR setup was applied to a DNA/RNA UV-cleaner UVT-S (BioSan, Latvia) PCR hood with a BioSan Combispin FVL-2400N vortex (BioSan, Latvia).

PCR was carried out with the “COMPLEX DENTOSCRIN” batch number 05/707Q/22 (OOO “Liteh”, Moscow, Russia) reagent kit for the periodontopathogenic bacterial DNA extraction using the PCR method.

A successful PCR analysis can be defined as when the amplification curves correspond to the internal control of the samples. A successful internal control was found at all study points, as shown in Figure 4. Therefore, the PCR results can be approved as consistent for the study of microorganisms’ presence in the samples.

For the statistical processing, negative PCR analysis results were expressed as “0”, and positive results were expressed quantitatively in genome equivalents in ml (GE/mL). For the estimation of “red complex” bacteria expressiveness in the oral microflora, the bacterial load of *T. denticola*, *T. forsythensis* and *P. gingivalis* was assessed as a summary of their genome equivalents in ml (GE/mL).

### 2.5. Saliva Biochemical Analysis

The metabolites of anaerobic and aerobic bacteria are represented by short-chain fatty acids (SCFAs) (C2–C6). For the qualitative and quantitative assessment of SCFA structure, saliva biochemical analysis was performed. Gas–liquid chromatography of volatile fatty acids (acetic, propionic, butyric, isobutyric, izovalerianic and isocaproic acids) was carried out on an AU 5800 automated biochemical analyzer (Beckman Coulte, Brea, CA, USA) with one ion-selective (ISE) and two photometric modules. Identification and quantitative assessment of the volatile fatty acid concentrations were conducted with the help of analytical references and the “MultiChrom” software system for chromatographic data processing. The total content ratio of acids with branched chains to acids with unbranched chains was assessed, and the ratio of propionic and butyric acids to acetic acid was considered to be the anaerobic index (C2–C4).

### 2.6. Statistical Data Processing

Statistical data processing was performed using the Statistica 12 software (StatSoft, Inc., Tulsa, OK, USA) with the application of “MANOVA” comprehensive analysis. The Mann–Whitney test was used for small samples. For the estimation of correlations, a nonparametric test (Spearman’s rank correlation) was applied.

## 3. Results and Discussion

The Silness–Loe hygiene index value among the patients varied from 0.3 to 1.8. Half of the patients had a good (14.6%) or acceptable (35.4%) oral hygiene status, while the others had unsatisfactory (45.8%) or poor (4.2%) oral hygiene. Therefore, according to the quality of oral hygiene, patients were equally represented in the sample.

### 3.1. Comparison of PCR Analysis, Proteolytic Activity Express Diagnostics and Saliva Biochemical Analysis Results

The results of the PCR analysis, proteolytic activity express diagnostics and gas–liquid chromatography analysis were analyzed. A significant strong correlation (r = 0.785, *p* < 0.001) between the PCR analysis results and the proteolytic activity of the bacteria of the interdental plaque was found.

In ten patients who underwent all three analyses, notable negative correlations between the values of the express device and the anaerobic index (r = −0.647, *p* < 0.01), the ratio of SCFA isomers to the total amount of acids (isoCn/Cn) (r = −0.639, *p* < 0.01) and the ratio of isoCn/Cn to the quantitative characteristics of the PCR analysis (r = −0.609, *p* < 0.01) were identified. These notable negative correlations indicate that the anaerobic index and the ratio of SCFA isomers to the total content of acids decrease with the accumulation of proteolytic bacteria in the oral cavity. The total “red complex” bacterial load had a high significant correlation with the values of the express device (r = 0.710, *p* < 0.01). Therefore, the results of all methods correlate with each other. The reason for this is that the highest “red complex” bacterial load (detected by the PCR analysis) demonstrates notable proteolytic activity (detected by the express diagnostic method) [25,28], and the 18–20% propionic acid, 10–12% butyric acid and 5.9–6.5% isoacid concentrations imply the growth of oral anaerobic microflora with proteolytic and hemolytic activity [29,30].

According to the significant correlation between the saliva biochemical analysis and proteolytic activity express diagnostics device data, it can be assumed that there is a connection between the presence of the “red complex” bacteria and the volatile fatty acid concentrations. The connection between volatile fatty acids and oral periodontopathogenic bacteria is under active investigation [29,30]. Moreover, volatile fatty acid analysis and accurate identification of their concentrations reflect more on the presence of *P. gingivalis* and *T. denticola* as opposed to all the “red complex” bacteria. Gas–liquid analysis requires expensive equipment and sample preparation and can be time-consuming; thus, it can be recommended as an additional diagnostic method, but not as an express diagnostic method for bacterial diseases of the oral cavity.

### 3.2. Comparison of PCR Analysis and Proteolytic Activity Express Diagnostics Results

Comparison of the results of the PCR analysis and express diagnostics of proteolytic activity was carried out for all 48 patients. The amplification curves, the total “red complex” bacterial load in GE/mL and the results of the express diagnostics of the interdental space samples of patients d19, d05, d01 and d10 are shown in Figure 5. In the d19 sample, all three representatives of the “red complex” bacteria were present. Their total amount was much higher compared to the other samples. Additionally, the blue spot on the chromogenic substrate layer that appeared after the application of this sample had a larger size and higher intensity.

For samples d05 and d01, the well-noted blue spots correspond to a high “red complex” bacterial content (three bacteria were present in the d05 sample at an amount of 2.7 × 10⁶ GE/mL, while in the d01 sample, only *T. forsythensis* was present at an amount of 1.5 × 10⁵ GE/mL).

For the d10 sample, no color spots were noticed, and only single bacterial cells of *T. forsythensis* were defined at an amount of 1.4 × 10^2^ GE/mL, which is below the threshold level of the express diagnostic method.

Therefore, the results obtained by the PCR analysis are comparable with the results of the express diagnostic method, indicating that the express diagnostic method can be applied for the detection of the “red complex” bacteria in biological samples.

### 3.3. Sensitivity and Specificity of the Express Diagnostic Method

After the comparison of the data obtained by the express diagnostics of proteolytic activity with the PCR analysis results, the threshold value of “red complex” bacterial DNA was 500 GE/mL. If the value is beyond 500 GE/mL, then the express method shows positive results and vice versa. The groups with positive and negative results statistically significantly differed in the “red complex” bacterial load (*p* < 0.01): the median value for the group with a positive device result was 3.4 × 10⁴ GE/mL, while the median value was 0 GE/mL for the other group.

For the study sample, the express diagnostic method for proteolytic activity demonstrated the diagnostic characteristics presented in Table 1.

### 3.4. Express Diagnostic Method Development

#### 3.4.1. Justification of the Sampling Location

The analysis of biomaterials collected from the interdental space and the back of the tongue demonstrated that the average amount of “red complex” bacterial DNA from the interdental space was significantly higher than that from the tongue (5.6% of interdental space bacteria). At the same time, a notable correlation between these two values was determined both in the average amount of the “red complex” bacterial mass (r = 0.51, *p* < 0.001) and PCR analysis results for the samples from the interdental space and the back of the tongue for *P. gingivalis* (r = 0.639, *p* < 0.001) and *T. denticola* (r = 0.58, *p* < 0.001). Therefore, both samples had a similar species spectrum, although they differed in the “red complex” bacteria concentration. These results are in agreement with the hypothesis of a simultaneous vegetation of periodontal pathogens in pathological gingival pockets and on the tongue mucosa [31]. Additionally, a significant correlation between the device values and the quantitative PCR analysis results of the interdental space samples was determined (r = 0.680, *p* < 0.001). This value was higher than the result for the tongue samples (r = 0.428, *p* < 0.001). The obtained data prove that the “red complex” anaerobic bacteria are mostly located in the anaerobic interdental space.

The highest correlation was estimated between the device results and the *T. denticola* content in the interdental space (r = 0.862, *p* < 0.001) and at the back of the tongue (r = 0.899, *p* < 0.001), which proves that this exact “red complex” bacterium was prevalent in the samples (83.1%).

Therefore, it is suggested that samples be collected from the interdental space for PCR analysis and assessment of microflora proteolytic activity.

#### 3.4.2. Incubation in the Process of Express Diagnostics of Proteolytic Activity in the Oral Cavity

To receive a reliable analytical signal and reproducible results in the proteolytic activity assessment, the external factors that impact the enzymatic reaction underlying the express diagnostic method should be considered.

It is well-known that the rate and temperature of enzymatic reactions are closely interrelated. As the temperature decreases, the proteolytic activity and the reaction rate decrease; as a result, the exposure time for the express diagnostics of proteolytic activity increases. The temperature rise induces irreversible protease denaturation that leads to a decrease in the biological activity and reaction rate. The exposure time also increases, which means a reliable result can be obtained. The choice of the optimal temperature conditions and the corresponding exposure time provides accuracy in the rate of proteolytic activity using the express diagnostic method.

It has been empirically demonstrated that the temperature range allowing optimal proteolysis of a substrate that is applied on the dry carrier of an express device varies from 36 to 45 °C, allowing a reliable and interpretable analytical signal to be obtained (Figure 6a). Similar data were also reported in previous work [32].

#### 3.4.3. Exposure Time in Express Diagnostics of Proteolytic Activity

In the laboratory study, it was proved that trypsin control solutions with concentrations ranging from 0.0167 mg/mL to 5 mg/mL applied on the dry carrier express device led to the appearance of the blue spot within 5 to 10 min, with the temperature range in the incubator maintained at 36 to 45 °C.

In this period, the color spot intensity and size increased, which was important for the unambiguous interpretation of the qualitative express device results. By the 5th minute in 25% of the cases, the level of chromogenic substrate hydrolysis did not reach the condition for a clear interpretation of the results according to the color change. Ten minutes after the application, color spots appeared in 100% of cases among the positive samples (Figure 6b). At the exposure time from 15 to 30 min, the spot intensity slightly increased, while the qualitative interpretation did not change in any sample.

In the clinical assay, the result of the express diagnostics of proteolytic activity obtained after a 10 min exposure time was proved to have a significant correlation with the PCR analysis results (r = 0.785, *p* < 0.001). The limited number of saliva results does not allow statistically accurate conclusions to be drawn at this stage, but a trend can be seen that requires further research on larger samples.

The correlations between the results obtained using the above-mentioned methods enable us to recommend the method that is the simplest in its implementation and interpretation of the results in clinical practice, and that can obtain the results in a short amount of time.

The limitations of this study relate to the sampling technique (sample location), temperature and incubation time, which can affect the interpretation of the results. These limitations necessitate strict adherence to the express diagnostic methodology for the proteolytic activity of periodontopathogens.

## 4. Conclusions

Based on this clinical research, it is recommended to use the new express method to register the proteolytic activity of the “red complex” of periodontopathogenic microorganisms. We recommend collecting a sample from the interdental space in the lateral group of teeth. In express diagnostics, the incubator temperature should be maintained between 36 °C and 45 °C for 10 min to achieve the optimal conditions for the enzymatic reaction.

The evidence for the effectiveness of the new diagnostic method is based on the correlation between the express diagnostics results and the PCR analysis used as the reference method (significant correlation, r = 0.785, *p* < 0.001).

Express diagnostics of the proteolytic activity of oral cavity bacteria demonstrated the following notable diagnostic characteristics: the method sensitivity and specificity were 0.875 and 0.928, respectively, as shown in the comparative study of the results from the PCR analysis. Summing up, the developed express diagnostic method can be valuable as a PD screening method that can be supplemented by PCR and saliva biochemical analyses. The express diagnostic method does not require expensive equipment or special training of the medical staff; thus, it can be applied widely in clinical practice.

## Figures and Tables

**Figure 1 dentistry-10-00217-f001:**
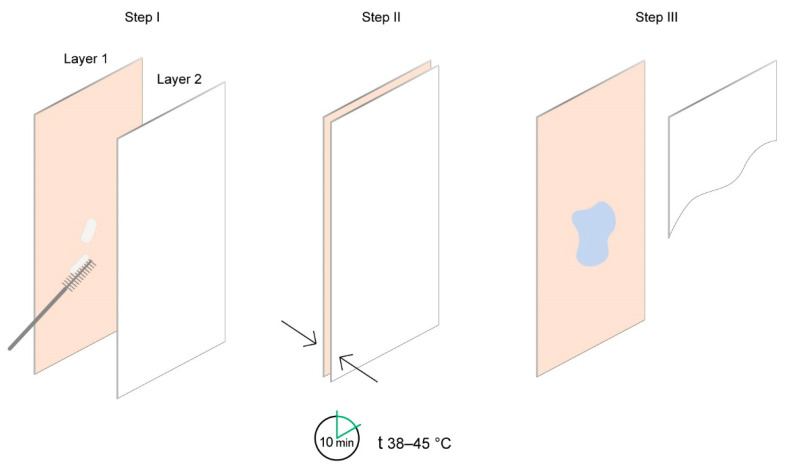
The scheme of the device layers for the express diagnostics of the proteolytic activity in the oral cavity: (Layer 1) the chromogenic test substrate; (Layer 2) the element containing color development dyes.

**Figure 2 dentistry-10-00217-f002:**
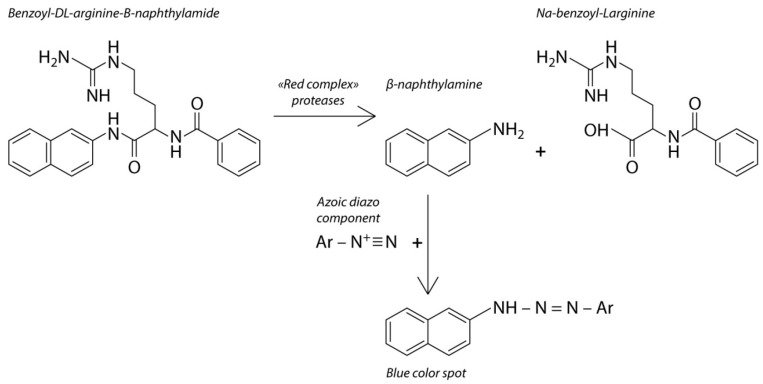
The reaction of chromophore formation that is the basis of the express diagnostics of proteolytic activity.

**Figure 3 dentistry-10-00217-f003:**
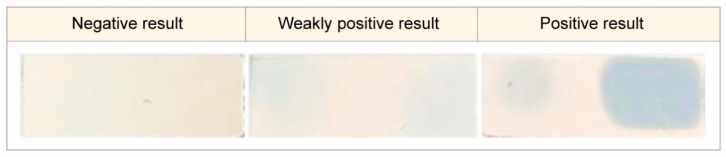
The color change scale for the proteolytic activity determination in the oral cavity.

**Figure 4 dentistry-10-00217-f004:**
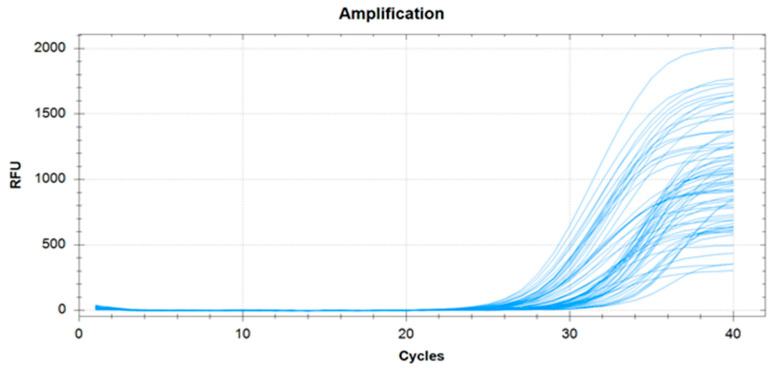
Amplification curves of the internal control in the biological samples.

**Figure 5 dentistry-10-00217-f005:**
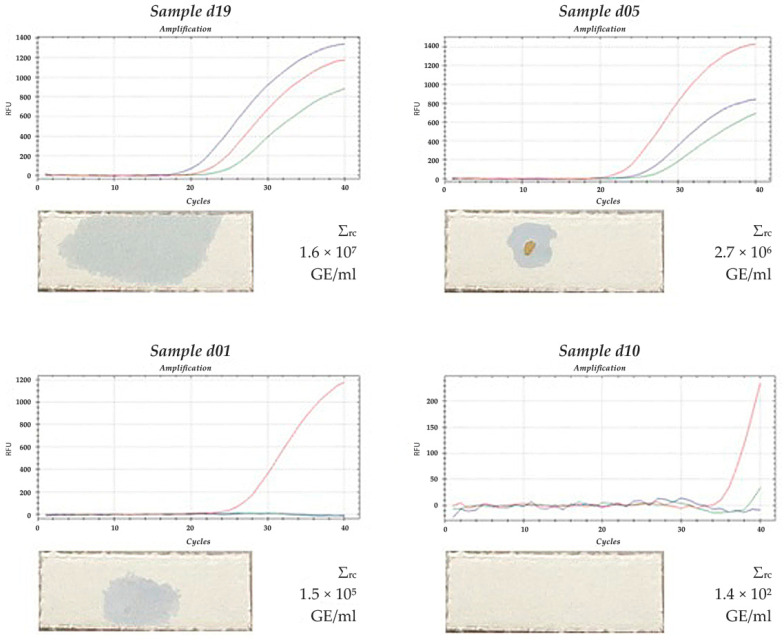
Comparison of the amplification curves and the results of the express diagnostics of the interdental space samples for patients **d19**, **d05**, **d01** and **d10**. PCR diagrams: the *Porphyromonas gingivalis* DNA amplification curve is blue; the *Tannerella forsythensis* DNA amplification curve is red; the *Treponema denticola* DNA amplification curve is green. Σrc—the total “red complex” bacterial load in genome equivalents in ml.

**Figure 6 dentistry-10-00217-f006:**
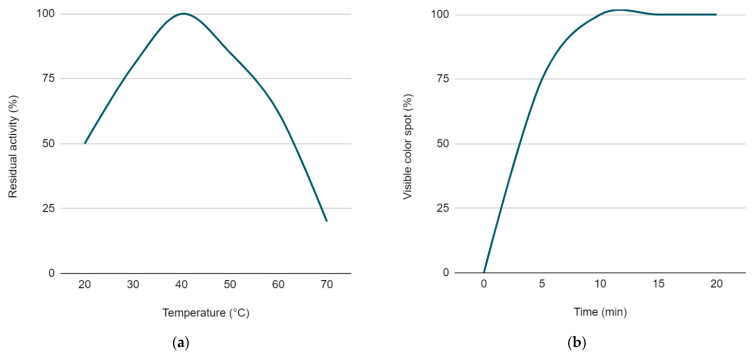
(**a**) Trypsin-like enzyme activity dependence on the temperature conditions. (**b**) The amount of proteolytic activity detected by the device when the color spot appeared after the trypsin solution application at 5, 10, 15 and 20 min.

**Table 1 dentistry-10-00217-t001:** Diagnostic characteristics of the express detection of proteolytic activity in the oral cavity.

Sensitivity	0.875
Specificity	0.928
False positive rate	0.071
False negative rate	0.125
Positive predictive value	0.954
Negative predictive value	0.812
Diagnostic accuracy	0.894

## Data Availability

No new data were created or analyzed in this study. Data sharing is not applicable to this article.

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
