# Peer review of "Express Diagnostics of Proteolytic Activity of Periodontopathogens—Methodological Approach"

_dentistry, 2022, doi:10.3390/dj10110217_

Round 1
Reviewer 1 Report
This study evaluates and propose a more commode and less expensive microbiota targeted alternative of screening periodontal lesions than PCR and saliva biochemical analyses. I appreciate the opportunity of reading the research, and I hope that my observations could be of use in improving the quality of the manuscript.
Introduction
Row 50: since the authors mentioned peri-implantitis caused by the microbial film aggressiveness and poor oral hygiene associated with orthodontic treatment facilitating the raise of microbial colonization, they should also referee to other oral conditions caused or implicated in the increase of the microbiota.
Row 69: the abbreviations should be put in parentheses after the written-out form as mentioned in “Instructions to authors” section.
Row 76: the paragraph does not have any reference. The last proposition is best suited in the next paragraph.
Material and Methods
Row 88: how were the patients recruited – inclusion and exclusion criteria?
It is not very clear how were the groups of the subjects formed.
Row 93: the Silness-Loe hygiene index values should be described in the “Results” section.
Results and discussion
The assays results should be correlated with the clinical results.
A phrase related to the limitations of the study could be introduced.
Author Response
Response to Reviewer 1 Comments
This study evaluates and propose a more commode and less expensive microbiota targeted alternative of screening periodontal lesions than PCR and saliva biochemical analyses. I appreciate the opportunity of reading the research, and I hope that my observations could be of use in improving the quality of the manuscript.
Thank you for your review and your valuable comments. We tried to take all your recommendations in improving quality of the manuscript. We are very pleased that you have read our manuscript, appreciated our work and left your professional feedback.
Moderate changes were made to the English language.
Introduction
Row 50: since the authors mentioned peri-implantitis caused by the microbial film aggressiveness and poor oral hygiene associated with orthodontic treatment facilitating the raise of microbial colonization, they should also referee to other oral conditions caused or implicated in the increase of the microbiota.
We have supplemented the text and indicated oral conditions caused by or associated with an increase in the microbiota.
Additionally, poor oral hygiene through the period of orthodontic treatment may facilitate the formation of dental plaque or tartar and lead to intense microbial colonization of the oral cavity. *The commensal microbiome has an important role in the maintenance of oral and systemic health. Balancing interference causes oral pathologies such as cavities, endodontic disease, periodontal disease, osteitis and tonsillitis and can be associated with the development of several systemic diseases, such as cardiovascular disease, ictus, pre-term childbirth, diabetes, pneumonia, obesity, colon carcinoma and psychiatric issues*.
Row 69: the abbreviations should be put in parentheses after the written-out form as mentioned in “Instructions to authors” section.
The abbreviations have been placed in parentheses after the written-out form.
*…transform such cytokines as tumor necrosis factor alpha (TNF-α), interferon gamma (IFN-γ), interleukin-6 (IL-6) and interleukin-8 (IL-8)… *
Row 76: the paragraph does not have any reference. The last proposition is best suited in the next paragraph.
Reference has been added. The last proposition was placed to the next paragraph. Additional corrections have been made to the text.
*At present, there are numerous methods for PD diagnostics: bacteriological analysis, the polymerase chain reaction (PCR) method, saliva gas–liquid chromatography and spectrophotometry, pH measurement and proteolytic activity evaluation [8].
The proteolytic activity of the “red complex” bacteria is assumed to be an important virulence factor; therefore, evaluation of the enzymatic activity of these bacteria allows for the prediction of inflammation in periodontal tissues at an early stage. Point-of-care testing for proteolytic activity offers many advantages as it is easily obtainable, sample collection is non-invasive and the procedure is performed in a doctor’s office. We hypothesized that express diagnostics of the proteolytic activity of periodontopathogens can help in determining the risk and severity of periodontal inflammation in periodontitis patients and can be used as a PD screening method. The aim of this study was to develop an innovative express methodology that allows for the estimation of the proteolytic activity of periodontopathogenic bacteria in dental and tongue plaque and to compare this method with molecular genetics methods, such as microbiological assays and saliva biochemical analysis.*
Material and Methods
Row 88: how were the patients recruited – inclusion and exclusion criteria?
It is not very clear how were the groups of the subjects formed.
The description of inclusion and exclusion criteria has been added.
*This study included patients with chronic gingivitis, chronic periodontitis and halitosis, as well as patients with soft and mineralized plaque without any complaints*. Before the procedure, consent was obtained from every patient. *The criteria for patient exclusion were: genetic diseases, acute and chronic somatic diseases in the decompensation stage, autoimmune diseases, acute and chronic infectious diseases and diseases of the oral mucosa*.
Row 93: the Silness-Loe hygiene index values should be described in the “Results” section.
The data on the results of the index plaque determination has been moved from materials and methods and described in the Results section.
Results and discussion
The assays results should be correlated with the clinical results.
A phrase related to the limitations of the study could be introduced.
Assays results have been supplemented. Limitations of the study were added.
*In the clinical assay, the result of the express diagnostics of proteolytic activity obtained after a 10-minute exposure time was proved to have a significant correlation with the PCR analysis results (r = 0.785, p < 0.001). The limited number of saliva results does not allow statistically accurate conclusions to be drawn at this stage, but a trend can be seen that requires further research on larger samples.
The correlations between the results obtained using the above-mentioned methods enable us to recommend the method that is the simplest in its implementation and interpretation of the results in clinical practice, and that can obtain the results in a short amount of time.
The limitations of this study relate to the sampling technique (sample location), temperature and incubation time, which can affect the interpretation of the results. These limitations necessitate strict adherence to the express diagnostic methodology for the proteolytic activity of periodontopathogens.*
Reviewer 2 Report
It is a relevant and novel topic, I suggest adding in background the justification of this new technique as well as some background of it. In addition, to emphasize the clinical importance that this new diagnostic technique could generate.
I suggest checking the general wording and spelling.
The main question was whether an innovative express estimation methodology of dental and tongue plaque periodontopathogenic bacteria proteolytic activity is effective in comparison with the molecular-genetic method of microbiological assay and the saliva 84 biochemical analysis.
The present investigation is relevant for clinical and accurate periodontal diagnosis. Periodontal disease is one of the most prevalent oral diseases in the world.
It is a novel diagnostic method, which has found a sensitivity of 0.875 and a specificity of 0.928 compared to the gold standard PCR.
It is a document that is easy to read and understand.
The conclusions are relevant to the clinical diagnosis of periodontal disease.
Author Response
Response to Reviewer 2 Comments
It is a relevant and novel topic, I suggest adding in background the justification of this new technique as well as some background of it. In addition, to emphasize the clinical importance that this new diagnostic technique could generate.
Thank you for your review and helpful suggestions. We appreciate your work. We are very pleased that you have read our manuscript.
Additional background and corrections have been made to the text.
*At present, there are numerous methods for PD diagnostics: bacteriological analysis, the polymerase chain reaction (PCR) method, saliva gas–liquid chromatography and spectrophotometry, pH measurement and proteolytic activity evaluation [8].
The proteolytic activity of the “red complex” bacteria is assumed to be an important virulence factor; therefore, evaluation of the enzymatic activity of these bacteria allows for the prediction of inflammation in periodontal tissues at an early stage. Point-of-care testing for proteolytic activity offers many advantages as it is easily obtainable, sample collection is non-invasive and the procedure is performed in a doctor’s office. We hypothesized that express diagnostics of the proteolytic activity of periodontopathogens can help in determining the risk and severity of periodontal inflammation in periodontitis patients and can be used as a PD screening method. The aim of this study was to develop an innovative express methodology that allows for the estimation of the proteolytic activity of periodontopathogenic bacteria in dental and tongue plaque and to compare this method with molecular genetics methods, such as microbiological assays and saliva biochemical analysis.*
I suggest checking the general wording and spelling.
English language and style were checked and moderated.
The main question was whether an innovative express estimation methodology of dental and tongue plaque periodontopathogenic bacteria proteolytic activity is effective in comparison with the molecular-genetic method of microbiological assay and the saliva 84 biochemical analysis.
Assays results have been supplemented. Limitations of the study were added.
*In the clinical assay, the result of the express diagnostics of proteolytic activity obtained after a 10-minute exposure time was proved to have a significant correlation with the PCR analysis results (r = 0.785, p < 0.001). The limited number of saliva results does not allow statistically accurate conclusions to be drawn at this stage, but a trend can be seen that requires further research on larger samples.
The correlations between the results obtained using the above-mentioned methods enable us to recommend the method that is the simplest in its implementation and interpretation of the results in clinical practice, and that can obtain the results in a short amount of time.
The limitations of this study relate to the sampling technique (sample location), temperature and incubation time, which can affect the interpretation of the results. These limitations necessitate strict adherence to the express diagnostic methodology for the proteolytic activity of periodontopathogens.*
The present investigation is relevant for clinical and accurate periodontal diagnosis. Periodontal disease is one of the most prevalent oral diseases in the world.
It is a novel diagnostic method, which has found a sensitivity of 0.875 and a specificity of 0.928 compared to the gold standard PCR.
It is a document that is easy to read and understand.
Thank you for your appreciation of the manuscript.
The conclusions are relevant to the clinical diagnosis of periodontal disease.